# Saint Petersburg 3D: Creating a Large-Scale Hybrid Mobile LiDAR Point Cloud Dataset for Geospatial Applications

Sergey Lytkin [1,*], Vladimir Badenko [1], Alexander Fedotov [1], Konstantin Vinogradov [2], Anton Chervak [1], Yevgeny Milanov [1] and Dmitry Zotov [1]

[1] Laboratory "Modeling of Technological Processes and Design of Power Equipment", Peter the Great St. Petersburg Polytechnic University, St. Petersburg 195251, Russia; badenko_vl@spbstu.ru (V.B.); afedotov@spbstu.ru (A.F.); chervak.av@edu.spbstu.ru (A.C.); milanov_es@spbstu.ru (Y.M.); zotov_dk@spbstu.ru (D.Z.)
[2] Department of Cartography and Geoinformatics, Institute of Earth Sciences, Saint Petersburg State University, St. Petersburg 199034, Russia; k.vinogradov@spbu.ru
* Correspondence: lytkin_sa@spbstu.ru

**Abstract:** At the present time, many publicly available point cloud datasets exist, which are mainly focused on autonomous driving. The objective of this study is to develop a new large-scale mobile 3D LiDAR point cloud dataset for outdoor scene semantic segmentation tasks, which has a classification scheme suitable for geospatial applications. Our dataset (Saint Petersburg 3D) contains both real-world (34 million points) and synthetic (34 million points) subsets that were acquired using real and virtual sensors with the same characteristics. An original classification scheme is proposed that contains a set of 10 universal object categories into which any scene represented by dense outdoor mobile LiDAR point clouds can be divided. The evaluation procedure for semantic segmentation of point clouds for geospatial applications is described. An experiment with the Kernel Point Fully Convolution Neural Network model trained on the proposed dataset was carried out. We obtained an overall 92.56% mIoU, which demonstrates the high efficiency of using deep learning models for point cloud semantic segmentation for geospatial applications in accordance with the proposed classification scheme.

**Keywords:** LiDAR; point cloud; mobile laser scanning; dataset; semantic segmentation; scene understanding; deep learning

## 1. Introduction

Geospatial applications refer to the utilization of spatial data in various fields and disciplines that involve location-based analysis, mapping, and visualization. It encompasses a wide range of domains, which include but are not limited to cartography, urban planning, environmental monitoring, natural resource management, disaster response, and transportation logistics. Geospatial applications rely on spatially referenced data, such as point clouds, to understand and analyze the Earth's surface and its features.

With the rapid development of LiDAR technology, working with 3D digital representations of real-world objects in the form of point clouds or meshes has become a common practice among industry practitioners and academic researchers. A point cloud is a digital record of objects and scenes [1] represented by a large number of 3D points, usually with different densities. Point clouds are a widespread type of data used in robotics, remote sensing, architecture, the construction industry [2], autonomous vehiclea, perception systems [3], geospatial applications [4], digital heritage [5], and many other areas. Most often, point clouds are obtained using active sensors based on LiDAR technologies mounted on a static or mobile platform. Moreover, a point cloud can be obtained from images using photogrammetry technology [6], as well as via bathymetric survey with a sonar. There are three main types of laser scanning technologies: terrestrial, mobile, and airborne [7].

Point cloud processing is an active area of research [8]. There are basic processing tasks that are independent of the application. These tasks include bringing individual point clouds to a common coordinate system (point cloud registration), semantic scene understanding (LiDAR data classification, semantic segmentation, and object detection), and data vectorization (automated primitive extraction, manual, automated, or semi-automated 3D modeling and 2D drawing) [9–11]. Among others, there are two independent areas of research in the field of point cloud processing related to the semantic scene understanding task: semantic segmentation for an autonomous vehicle's perception [12] and LiDAR data classification for geospatial or remote sensing applications [13]. Despite the difference in terminology, the result is the same in both cases: we need to assign a special class label to each point [14]. Thus, the result of any of these procedures is a point-wise classification of the input data. Given the growing popularity and potential of deep learning models, the remote sensing community is trying to adapt existing relevant semantic segmentation artificial intelligence models for processing denser and more diverse point clouds while solving the task of LiDAR data classification, instead of using traditional rule-based heuristic algorithms [15].

The term LiDAR data classification is used by remote sensing specialists to solve the problem of determining the category of an object and assigning the corresponding additional attribute to each point obtained during lidar surveys. LiDAR data classification is a classic problem in the area of point cloud processing for geospatial applications, which originated with the start of the first airborne lidars operation used for landscape surveys. It should also be noted that the terms "point cloud classification" [8], "point cloud segmentation" [16], and "point cloud semantic segmentation" [17], which involve the task of assigning an additional attribute to each point, are often used interchangeably. In this article, we use the terminology that has historically developed in geospatial applications and is used in the processing of remote sensing data, namely the term "classification". Therefore, this term should not be confused with the similar term used in computer vision, which means to assign a special class label to the entire input point cloud, rather than to assign labels for each point of the input point cloud as in geospatial applications [14].

To point cloud classification, currently, the most promising approach, in our opinion, is the use of deep learning models. The availability of digital data and the increase in computing power in recent years have led to the development of deep learning in many areas, especially in computer vision [18]. For example, in the field of autonomous driving, semantic segmentation of point clouds has become an increasingly relevant research topic in recent years [19].

The application of deep learning models is preceded by a training process that requires large-scale point cloud datasets with high-quality annotation [13]. The advent of such large-scale image datasets, such as ImageNet [20], PASCAL VOC [21], Microsoft COCO [22], and others, contributed to significant progress in image processing using deep learning models [13]. The lack of a wide variety of similar datasets, which would be related to the point cloud datatype, significantly slows down the development of learning-based LiDAR data processing methods. Therefore, the development of new point cloud datasets is an important task in the development of such learning-based approaches.

The vast majority of existing point cloud datasets are designed for research on learning-based point cloud processing methods for real-time autonomous vehicles' perception of their environments. Such studies on LiDAR data classification for remote sensing data processing and geospatial applications are at the very beginning of their development stage [23,24]. Therefore, studies on the development of learning-based point cloud processing methods for geospatial applications use existing publicly available perception datasets to train and evaluate the performance of models. Spatial data and the purposes of their processing in autonomous driving and geospatial applications differ from each other. Therefore, a special analysis of existing point cloud datasets is required to assess the possibility of their application in the context of a geospatial application.

There are a huge number of classification schemes for different point cloud datasets, which makes it impossible to use such datasets together. To bridge this gap, the development of a unified classification scheme for geospatial applications is needed. Such a scheme should correspond to the goals of data processing in geospatial applications and include universal object categories into which any scene represented by dense outdoor mobile LiDAR point clouds can be divided. Based on such a classification scheme, it will be possible to develop unified datasets, which can be used as a single set of data for training deep learning models.

The objective of the research is to develop a new large-scale mobile 3D LiDAR point cloud dataset for outdoor scene semantic segmentation tasks, which has a set of universal classes for geospatial applications. The main contributions of our work are as follows:

- We provide an analysis of existing point cloud datasets (point cloud characteristics and classification schemes) in terms of their use in the context of geospatial applications;
- We propose a classification scheme that contains 10 object categories most suitable for geospatial applications, instead of the currently used object categories within existing classification schemes, which are more suitable for the perception of the environment using autonomous vehicles;
- We present our large-scale mobile point cloud dataset (available at: https://github. com/lytkinsa96/Saint-Petersburg-3D-Dataset, accessed on 10 April 2023), which consists of 34 million real-world points and 34 million synthetic points located along trajectories that are 0.7 and 2.1 km long, respectively;
- We present the results of the performance evaluation of the state-of-the-art deep learning model—Kernel Point Fully Convolutional Neural Network (KP-FCNN) [25]—trained on our dataset.

The structure of this paper is as follows: Section 2 provides a detailed analysis of existing mobile LiDAR point cloud datasets for semantic segmentation of outdoor scenes; Section 3 describes the methodology for creating our large-scale mobile LiDAR dataset—Saint Petersburg 3D (SP3D); Section 4 presents the performance evaluation for the state-of-the-art deep learning model (KP-FCNN) trained on our dataset; and Section 5 summarizes the results of the work and discusses possible future research directions.

## 2. Related Works

In this section, we provide a comprehensive analysis of existing point cloud semantic segmentation point cloud datasets, which consists of virtual (synthetic) and real outdoor scenes obtained using mobile mapping systems. Here, we do not consider other point cloud generation techniques, such as photogrammetry [26], RGB-D [27] or aerial laser scanning [28], since their fields of view, density distributions, effective ranges, and accuracies differ significantly from MLS (Mobile Laser Scanning) 3D LiDAR data. Moreover, photogrammetry approaches cannot effectively penetrate dense vegetation, which leads to insufficient description of the underlying surfaces. The aforementioned features affect the initial point cloud representation, which reduces the robustness of learning-based approaches. However, despite the distinctive features of TLS (Terrestrial Laser Scanning) and MLS, we also considered Semantic3D [13] as one of the few datasets mainly focused on spatial data processing in the context of geospatial applications and remote sensing. Among the mobile LiDAR point cloud datasets, we considered only those datasets whose LiDAR scans can be interpreted as dense point clouds. Thus, we did not consider multimodal datasets (including LiDAR data with addition to RADAR data or digital images), aimed at real-time perception of the environment, such as nuScenes [29], A2D2 [30], KITTI [31], ApolloScape [32], Lyft Level 5 [33], and Waymo OD [34].

### 2.1. Real World Data

Existing outdoor point cloud semantic segmentation datasets have a number of shortcomings, especially in the context of their use for geospatial applications, which will be discussed further.

Oakland3D [35] was obtained using the Navlab11 [36] mobile scanning system. There are five object categories: wires, poles, load bearing, facades, and others. Oakland3D has many occlusions and a low point density distribution. As a result, the number of points labeled as power lines or road signs is reduced, which greatly increases the class imbalance of the dataset.

The Paris-rue-Madame dataset [37] was acquired with the L3D2 [38] mobile scanning platform equipped with Velodyne HDL-32. Available classes include facades, ground, cars, motorcycles, pedestrians, traffic signs, and others. The described dataset is characterized by a high relative measurements error (local noise), a high point density, and annotation inaccuracy at the boundaries of objects. The scene has a low object variety and is mostly represented by points related to the ground, facades, and cars.

The iQmulus dataset [39] is acquired using a Stereopolis II [40] mobile scanning system, which is equipped with two Riegl LMS-Q120i and one Velodyne HDL-64E. Each point is associated with one of the following classes: ground, building, other surface, static object, dynamic object, natural object, other object, other, and unclassified. The described scene has a small variety of objects and is mainly represented by points associated with the ground, facades, and cars. Point cloud annotation was carried out using an interface displaying a point cloud in the sensory space, i.e., based on flat projections of the point cloud. It is known that this approach leads to annotation errors on the boundaries of objects because of the mutual registration of points and pixels. Semantic3D [13] contains terrestrial laser scanning data belonging to one of the following categories: man-made terrain, natural terrain, high vegetation, low vegetation, buildings, hard scape, scanning artefacts, and cars. There is also an additional label for unlabeled points, which is not used for training [13]. Each scene is acquired using several scan stations with small overlap, resulting in a lot of occlusions in the scenes. Terrestrial laser scanning resulted in a high point density near the scan stations and a low point density at the boundaries of the point clouds. A significant number of points in each scan are mainly represented by the ground surface and the scenes have an unbalanced class distribution. As a result, the training examples do not match the real-world scenes.

The Paris-Lille-3D dataset [41] obtained using the L3D2 mobile scanning platform [38] is equipped with Velodyne HDL-32E. The Paris-Lille-3D dataset includes the following 10 classes: unclassified, ground, building, pole, bollard, trash can, barrier, pedestrian, car, and natural. Paris-Lille-3D contains data from two different cities in France. Point clouds have a high point density and high-quality annotations and contain different class instances compared to previously published datasets.

The SemanticKITTI dataset [42] consists of annotated sequences derived from single Velodyne-HDL-64E scans of the original KITTI dataset [31]. It contains 28 object classes. SemanticKITTI is the largest existing public dataset for semantic segmentation of outdoor scenes; it has a high-quality annotation, but also has several disadvantages. For example, the dataset has low point density for open spaces and a highly detailed class tree, which leads to extremely unbalanced distribution of points between classes.

The Toronto-3D dataset [43] includes the following eight classes: road, road marking, natural, building, utility line, pole, car, and fence. The point clouds were acquired using a vehicle-mounted Teledyne Optech Maverick system. It is currently one of the most suitable datasets for dense LiDAR data classification for geospatial applications. However, the existence of "road markings" label reduces the performance of deep learning models. Moreover, the main drawback of Toronto-3D is poor labeling quality.

TUM-MLS-2016 dataset [44] was acquired using an experimental MODISSA platform equipped with two Velodyne HDL-64E sensors. The dataset has the following eight categories: man-made terrain, natural terrain, low vegetation, high vegetation, buildings, vehicles, scanning artifacts, and hard scape. The main feature of the TUM-MLS-2016 dataset is the presence of two LiDAR sensors, which provide sufficient coverage of the scene and high scanning density compared to most existing sets.

SemanticPOSS dataset [45] has 14 categories of objects, including people, cars, buildings, plants, road signs, trash cans, poles, and other objects. The point clouds were obtained using a mobile scanning system equipped with a Pandora sensor module, which has a 40-channel LiDAR with a vertical resolution of 0.33 degrees and a claimed range of up to 200 m.

Urban SGPCM dataset [46] has 31 object categories. Data were obtained using the Leica Pegasus: Two Ultimate mobile mapping system. Among all the previously described datasets, Urban SGPCM is the only one that consists of data obtained using a full–fledged mobile mapping platform.

### 2.2. Synthetic Data

Developing large-scale point cloud datasets requires special equipment for 3D data acquisition, as well as much more manual human effort to annotate it, compared to annotating 2D images. This fact is due to the strongly varying point density identified during laser scanning and difficulties in automated scene interpretation [13]. Moreover, such projects are expensive and thus inaccessible to most researchers. To address this problem, a certain number of works investigate the use of synthetic (data collected from a source other than the real world) LiDAR point cloud data to train modern deep learning models.

In recent years, the scientific community developed solutions for generating such synthetic data [47–51]. Moreover, there are several publicly available synthetic point cloud datasets [12,52–54] that will be discussed further. Here, we are only interested in the semantic segmentation of point cloud data; therefore, we do not consider object detection datasets that mimic the KITTY vision benchmark [31], such as GTA V [55] and PreSIL [56].

SynthCity [52] was generated through simulation of a mobile laser scanning platform equipped with Velodyne HDL-64E in a virtual environment, which is similar to a typical New York City and suburban territories simulation designed using BlenSor [47]. Dataset include the following nine categories: road, pavement, ground, natural ground, tree, building, pole-like, street furniture, and car. SynthCity was designed to stimulate the study of the ability of models to generalize from synthetic data to real world data. The SynLiDAR dataset [53] was aimed at research in the field of knowledge transfer from synthetic to real 3D data. It has 32 objects categories. To the best of our knowledge, SynLiDAR is the largest synthetic point cloud dataset in existence. However, it is intended to improve the perception of the outdoor environment for autonomous vehicles and, therefore, has relevant characteristics, such as an extremely detailed class tree and a limited field of view. The KITTI-CARLA [54] dataset consists of synthetic point clouds, which are generated via the CARLA [45] simulator through simulating the sensor with similar characteristics to the Velodyne HDL-64E, which is widely used in the autonomous driving industry. Each point belongs to one of the 23 object classes. This dataset is characterized by excessive density, which significantly increases the number of points and the amount of data. The unrealistic virtual environment, which is unusual for real-world scenes, should also be noted. The Paris-CARLA-3D hybrid dataset [12] augments the KITTY-CARLA dataset with real data acquired via a mobile scanning system equipped with Velodyne HDL-64E. To the best of our knowledge, Paris-CARLA-3D is the only hybrid dataset that consists of both real-world and synthetic point clouds. However, there is one huge drawback, which is the difference in representation of real-world and synthetic point clouds of vegetation. This problem occurs because standard CARLA maps approximate complex geometries using low-poly meshes with an insufficient number of vertices.

### 2.3. Summary

The point clouds used in geospatial applications differ significantly from the point clouds used in autonomous driving. These differences include field of view; measurement range; noise level, which is affected by the accuracy of LiDAR sensors; and IMU/GNSS equipment. In autonomous driving, the input data are represented by a single LiDAR scan centered on the vehicle, and semantic segmentation is performed in real time using

neural networks [12]. Here, in using the term 'scan', we consider a sparse point cloud obtained as a result of laser scanning for a fixed period of time (usually 1 s). In geospatial applications, the input data are represented by dense large-scale point clouds, and the classification is performed without time limits at the post-processing stage using rule-based heuristic algorithms. Thus, the point clouds used in LiDAR data classification are significantly different from the raw data used in autonomous driving systems. The spatial data used in geospatial applications are subject to a number of specific requirements that depend on the scale of the output and are regulated using regulatory documents. Such requirements include the accuracy of determining the coordinates of points, point density, and point spacing.

An analysis of the classification schemes used in existing public datasets shows that semantic segmentation of autonomous vehicles' perception abilities is mainly focused on the identification of static and dynamic instances, as well as small objects, such as road signs, traffic lights, curbs, lane markings, and other items, for further decision-making. There are several datasets [12,37,41,43], which are suitable for creating high-definition maps (HD maps), that are similar to typical dense point clouds used in geospatial applications; however, their classification schemes still contain the labels inherent in perception datasets. On the other hand, LiDAR data classification for geospatial applications are aimed at input point cloud subdivision into certain parts, which is related to larger objects, such as ground surface, vegetation, buildings, power lines, and other objects, to further generate accurate 3D models of the environment [9].

Existing point cloud datasets were extensively analyzed to assess their suitability for geospatial applications. However, it is evident that their utility in this context is largely constrained. While these datasets can be effectively employed for the identification of ground points due to the inclusion of a 'Ground' class, their limitations become apparent when considering the automation of topographic sheet creation or 3D modeling based on LiDAR survey results.

One of the primary limitations arises from the inherent characteristics of publicly available datasets, which are predominantly generated using close-range LiDAR systems. As a result, the accuracy, range, and density of measurements in these datasets significantly differ from those obtained using full-fledged mobile mapping systems. Consequently, their use in geospatial applications, which requires precise and comprehensive measurements, is restricted due to the inadequate LiDAR characteristics presented in these datasets.

Moreover, the existing classification schemes employed in these datasets primarily focus on addressing the requirements of autonomous driving applications, emphasizing the distinction between static and dynamic objects. However, for geospatial applications, the key objects of interest typically include fences, poles, buildings, and similar static structures. Unfortunately, the available datasets do not provide the necessary classification scheme for accurately identifying and extracting these features. Consequently, the automation of topographic sheet creation or 3D modeling based on LiDAR point clouds using existing datasets remains unfeasible.

In summary, the analysis of existing point cloud datasets reveals their limited suitability for geospatial applications. The differences in LiDAR characteristics and classification schemes employed in these datasets hinder their effective utilization in automating topographic sheet creation and 3D modeling. To overcome these limitations, the development of new point cloud datasets specifically designed for geospatial applications is crucial to enable accurate and automated topographic surveying processes and 3D modeling.

The reviewed datasets are summarized in Table 1.

**Table 1.** Point cloud datasets for semantic segmentation of outdoor scenes.

| LS Type | Type | Dataset (Year) | Length, km. Area, km² | Number of Classes | Number of Points (mln) |
|---|---|---|---|---|---|
| TLS | | Semantic3D [13] (2017) | - | 7 | 4000 |
| | | Oakland 3D [35] (2009) | 1.4 | 7 | 1.6 |
| | | Paris-rue-Madame [37] (2014) | 0.16 | 6 | 20 |
| | | iQmulus [39] (2015) | 10 | 8 | 300 |
| | Real-world | Paris-Lille-3D [41] (2018) | 2 | 10 | 143.1 |
| | | SemanticKITTI [42] (2019) | 39.2 | 25 | 4500 |
| | | Toronto 3D [43] (2020) | 1 | 8 | 78.3 |
| MLS | | TUM-MLS-2016 [44] (2020) | 1 | 8 | 40 |
| | | SemanticPOSS [45] (2020) | 1.5 | 14 | 216 |
| | | Urban SGPCM [46] (2022) | 21 | 30 | 11 |
| | | SynthCity [52] (2019) | - | 9 | 367.9 |
| | Synthetic | SynLiDAR [53] (2021) | - | 32 | 19,000 |
| | | KITTI-CARLA [54] (2021) | 5.8 | 23 | 4500 |
| | Hybrid | Paris-CARLA-3D [12] (2021) | 0.5 (5.8) | 21 | 60 (700) |
| | | SP3D (*ours*) | 0.7 (2.1) | 10 | 34 (34) |

## 3. Materials and Methods

In this section, we provide a description of the methodology for creating a SP3D dataset, which consists of the following steps:

1. Development of requirements for the dataset (Section 3.1);
2. Development of a dataset classification scheme (Section 3.2);
3. Data collection and annotation (Section 3.3);
4. Description of the dataset evaluation procedure (Section 3.4).

### 3.1. Development of the Requirements for the Dataset

1. When developing data sets for geospatial applications, it is necessary to use real data from full-fledged mapping systems, as well as synthetic data obtained as a result of simulating virtual sensors with similar characteristics. Velodyne HDL-64E is the most commonly used sensor among all reviewed datasets [12,39,42,45,52,54]. However, as of 2021, it was discontinued. Moreover, Velodyne HDL-64E-based mapping systems are primarily used for perception of the environment using autonomous vehicles. Such systems are inferior in characteristics to full-fledged mapping systems from manufacturers such as Riegl, Leica, Trimble, etc.

2. The label verification procedure should be considered, as well as the use of semi-automatic ground truth production approaches, to improve annotation quality and reduce labor costs. Some of the publicly available datasets [37,39,43] have low-quality annotations that can reduce the performance of deep learning models. The main reasons for the low-quality annotations are the human factor and reprojection errors that occur when point cloud labeling is performed based on 2D projections. Several studies investigate semi-automatic ground truth production, including point cloud segmentation, followed by manual refinement [37], and point label prediction based on deep learning model trained on similar dataset, which is also followed by manual refinement [46]. As a result, faster labeling is reported in [37,46].

3. It is necessary to use more suitable and complex 3D models of real objects for synthetic data generation. It is also possible to use triangulated meshes from real-world point clouds to represent such objects. One of the main drawbacks of several existing synthetic datasets [37,43] is the low realism of the geometry of individual objects in the virtual environment. This problem is due to the insufficient number of mesh vertices representing the objects of complex shape, such as trees and bushes. Simulation of scanning of such objects leads to significant distortions in the resulting geometry

relative to real-world prototypes. The low results of transfer learning from synthetic to real-world data confirm this assumption [12].

4. The development of SP3D dataset should be based on hybrid data containing both real and synthetic point clouds, which are preferable due to the possibility of their use in transfer learning from synthetic to real point clouds studies. There is a domain gap between real-world and synthetic point clouds [53]. This domain gap is raised from the differences between real-world and virtual scene layouts, as well as from the specific realism of the 3D geometry used in synthetic scenes. As the point cloud acquisition is performed in a more controlled way than image acquisition, we assume that point cloud datasets for geospatial applications can be augmented with synthetic data. Synthetic data should be derived from virtual scenes with realistic 3D geometry and layouts similar to the features found in real-world scenes. The simulation must be performed using a virtual sensor that has the characteristics of the real one. This method is possible due to several conceptual assumptions. Semantic segmentation is one of the steps in LiDAR data classification for geospatial applications, which usually includes filtering steps. The use of deep learning models often implies data pre-processing, including optimization steps, which also distorts the initial data fed to the input of the artificial neural network. As far as we know, there is only one study that previously used synthetic data to augment real-world point clouds [12].

### 3.2. Development of a Dataset Classification Scheme

The analysis of related works showed that existing point cloud datasets had different classification schemes, which made it impossible to use them together for training and testing deep learning models. We believe that in order to develop a truly large-scale dataset suitable for use in the context of geospatial applications, it is first necessary to develop a universal classification scheme. Thus, through developing synthetic and real point cloud datasets that followed the universal classification scheme, it became possible to share single datasets for training and testing deep learning models capable of consistently generalizing new, unseen scenes.

In 2003, the American Society for Photogrammetry and Remote Sensing (ASPRS) published a standard file format for storing laser scanning data (las-format) [57] and required its implementation in the US landscape scanning and construction software market. Currently, in the field of geospatial applications, developers of point cloud classification programs strive to form classification schemes that are compatible with the ASPRS standard (usually null and first 12 classes). These schemes include:

0. Created, never classified (the point was not subjected to the classification process);
1. Unclassified (undetermined state of a point in the classification process);
2. Ground;
3. Low Vegetation;
4. Medium Vegetation;
5. High Vegetation;
6. Building;
7. Low Point (noise);
8. Model Key Point (mass point);
9. Water;
10. Reserved for ASPRS Definition;
11. Reserved for ASPRS Definition;
12. Overlap Points.

The above classification scheme had disadvantages for deep learning models. For example, for the classification of vegetation of different heights, as well as various landscape coverages, the use of neural networks is less effective than the use of heuristic algorithms. In addition, the ASPRS schema did not take into account other objects specific to survey applications, such as fences, power lines, street lighting poles, etc. This classification scheme also did not take into account objects that, although not of interest from the point

of view of remote sensing, are present in scenes in large numbers. These objects included vehicles, improvement objects, pedestrians, etc.

Considering the specifics of the neural network's operation, as well as the ultimate goals pursued in geospatial applications, we believed that the task of LiDAR data classification could be solved through jointly using heuristic algorithms and deep learning models. When developing a classification scheme for a point cloud dataset, we took into account the fact that in the context of geospatial applications, the semantic segmentation of dense large-scale point clouds is only one of the stages involved in their processing. Therefore, when developing a classification scheme, we sought to select classes that, on the one hand, would allow us to divide the scene into the smallest number of the most common object categories, and on the other hand, could subsequently be divided into smaller categories using existing heuristic algorithms or alternative methods. The developed classification scheme consisted of the following 10 categories of objects:

0. *Unclassified*: Any point that did not belong to any of the following nine categories (undetermined state of a point in the classification process, analogue of the "Unclassified" class of the ASPRS specification);
1. *Ground*: Any ground surface, including paved roads, sidewalks, and natural terrain;
2. *Building*: All parts of multi-story buildings, including tenements, shopping malls, etc.;
3. *Pole*: Pole-like objects, including traffic signs, lamp posts, and power line poles;
4. *Fence*: Any vertical barrier, including metal or wooden fence, brick walls, etc.;
5. *Utility lines*: All utility lines between poles or buildings;
6. *Artifact*: Outlier points and points of moving objects, such as pedestrians or vehicles;
7. *Vehicle*: All points belonging to static vehicles or their parts identified in motion;
8. *Natural*: Trees and bushes;
9. *Other object*: Clutter class, including bus stops, billboards, static pedestrians, etc.

### 3.3. Data Collection and Annotation

3.3.1. Real-World Data Collection

Real-world data were collected on Komsomola Street near the Finlyandsky railway station in St. Petersburg, Russia. Unlike the perception datasets, we used a full-fledged Riegl VMX-450 mobile mapping system. It has two Riegl VQ-450 laser scanners, IMU and GNSS equipment, and 6 high-resolution digital cameras. Riegl VQ-450 has a pulse repetition rate of up to 550,000 measurements per second and a scan speed of up to 200 scans per second. Therefore, to reduce the volume and density of data, we processed measurements from only one lidar. The raw measurements were processed using the Riegl RiPROCESS software to create a final dense point cloud with RGB components assigned to each point.

Real-world point clouds from the dataset describe objects of a typical dense urban environment formed at the beginning of the 20th century. Real-world subset contains static objects represented by houses up to 50 m high (5 floors) with historical facades, fences, poles, power lines, and landscape objects (trees, advertisements, trash cans, benches, etc.). The data also contain many dynamic objects, such as pedestrians and moving vehicles. The listed features represent challenges for understanding 3D scenes.

Each point had the following additional attributes: RGB, ScanAngleRank, ScanDirectionFlag, NumberOfReturns, ReturnNumber, GpsTime, Intensity, Original_cloud_index, Amplitude, EdgeOfFlightLine. Moreover, each point was assigned one of the 10 object categories defined by the proposed classification scheme.

3.3.2. Real-World Data Annotation

For data annotation, we used an approach similar to those used in SemanticPOSS [45] and Urban SGPCM [46], which used pre-trained models for the initial coarse annotation. Figure 1 illustrates a step-by-step data annotation procedure.

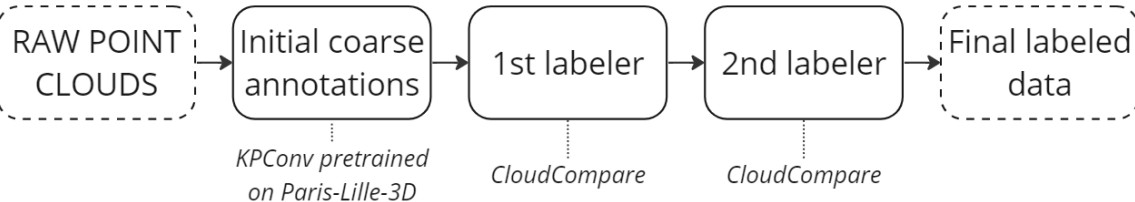

**Figure 1.** Step-by-step data annotation procedure.

### 3.3.3. Synthetic Data Collection

The synthetic part of our SP3D dataset was generated from 3 realistic virtual environments of urban areas constructed from open-source 3D models using Blender software. To reduce the domain gap when creating 3D models, we tried to use objects that had realistic geometry and place them inside the scenes in such a way that the scene layouts correspond to the real world. Generation was performed using the HELIOS++ software, which allowed us to simulate LiDAR sensors in virtual environments [58]. As in the Paris-CARLA-3D dataset [31], we also reproduced our real-world mobile mapping platform to perform this simulation. To resemble our real-world raw measurements processing pipeline, we also used one Riegl VQ-450 sensor in HELIOS++.

### *3.4. Description of the Dataset Evaluation Procedure*

### 3.4.1. Metrics

For point cloud semantic segmentation, deep learning models take a point cloud as input and a prediction of object label as output for each point [14]. As in similar works, to evaluate the performance per class c, we used the well-known Jaccard Index or the Intersection over Union (*IoU*) metric [21], as given by this formula:

$$IoU_C = \frac{TP_C}{TP_C + FP_C + FN_C} \tag{1}$$

where *TPc*, *FPc*, and *FNc* denote the number of true positive, false positive, and false negative class c predictions. We also provide mean *IoU* (*mIoU*), which is defined as the arithmetic mean of *IoU*, as given by the formula:

$$mIoU = \frac{1}{N} \sum_{i=1}^{N} IoU_C \tag{2}$$

where *N* is an overall number of classes.

In our experiment, we did not take into account the "Unclassified" and the "Artifacts" classes, because the "Unclassified" class is too diverse and serves as a container for points whose label predictions have a probability below a threshold value. Moreover, the "Artifacts" class contains outliers and point of moving objects, which are difficult to simulate. In addition, point cloud processing for geospatial applications includes a data filtering stage, which is precisely aimed at removing points belonging to the "Artifacts" class. Therefore, to evaluate the performance of the baseline method, we used 8 instead of 10 classes at the training and testing stages of our experiment.

### 3.4.2. Baseline Method

There are many different approaches to point cloud semantic segmentation, which are well described in [59,60]. We considered KP-FCNN model [25] as a baseline method, because it has high performance on modern benchmarks, such as Paris-Lille-3D [41] and SemanticKITTI [42]. It is based on U-net architecture and a state-of-the-art point convolution operator for semantic segmentation of dense large-scale point clouds, which is consistent with its high performance using most considered datasets. The principle of operation of KPConv is based on the use of 3D filters with weights spatially located in Euclidean space

over a small set of kernel points. Each kernel point has its own area of influence, which is determined based on the correlation function. The Deformable KPConv also learns local shifts in order to adapt the location of kernel points according to point cloud geometry [25].

KP-FCNN is a fully convolutional network for segmentation. The encoder part is a five-layer classification convolutional network, where each layer contains two convolutional blocks, with the first one being strided except for in the first layer. Convolutional blocks are designed like bottleneck ResNet blocks with a KPConv replacing the image convolution, batch normalization, and leaky ReLU activation. After the last layer, the features were aggregated via global average pooling and processed via the fully connected and softmax layers akin to the process used in an image CNN. Deformable KPConv only used deformable kernels in the final 5 KPConv blocks. The decoder part uses nearest upsampling to identify the final pointwise features. Skip links were used to pass the features between intermediate layers of the encoder and the decoder. Those features were concatenated to the upsampled features and processed through a unary convolution, which is the equivalent of a $1 \times 1$ convolution in image or a shared MLP in PointNet [25]. Architecture of KP-FCNN model is presented in Figure 2.

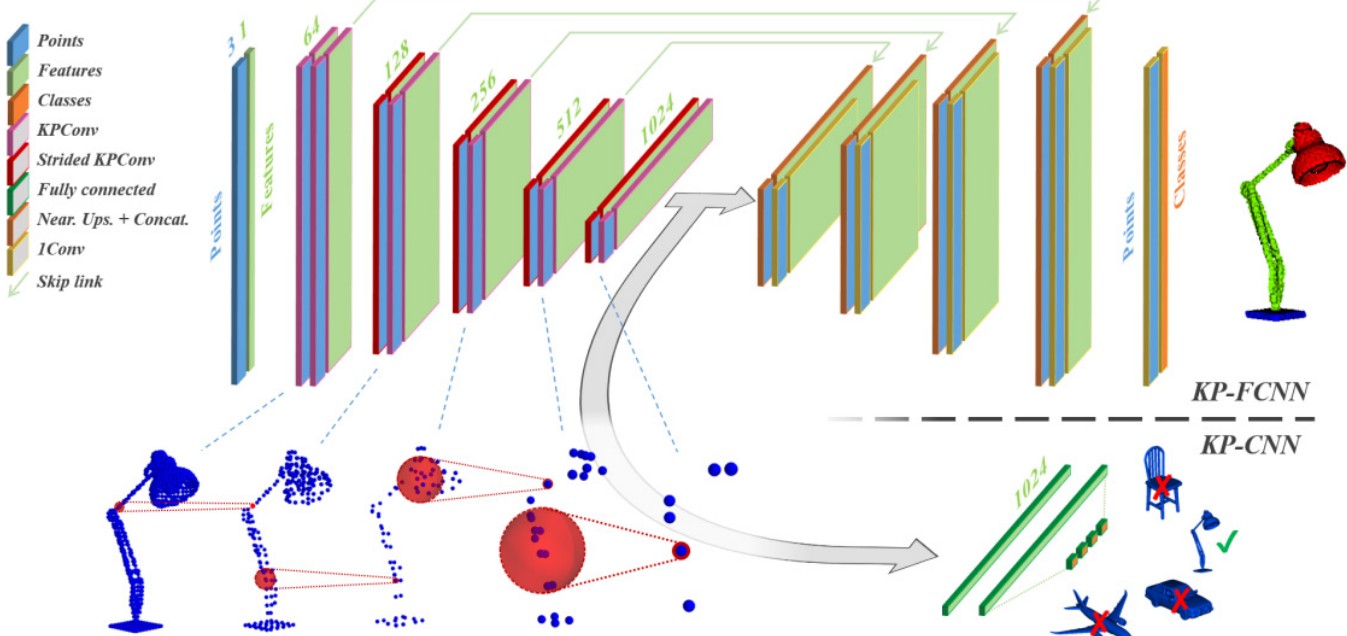

**Figure 2.** Illustration of KP-FCNN architecture [25], which was used as a baseline method in our experiments.

As mentioned in [12], dense large-scale outdoor point clouds cannot be stored in memory, due to the large number of points. For this reason, the input point clouds for semantic segmentation models were represented via subclouds of fixed form, including spheres of a given radius [14]. In our experiment, we used the same training procedure and hyperparameters used in the original paper [25] for Paris-Lille-3D and Semantic3D semantic segmentation. In particular, we denoted the input sphere radius as 3 m, while cell size (dl0) was 6 cm. The kernel points influence distance was set to be the same as the cell size at a certain network layer. The convolution radius was set as $r_j = 5 \times \mathrm{dl}_j$. The number of Kernel Points was set to 15. However, unlike the original paper, we do not add any features as input, except for a constant feature equal to 1, which encodes the geometry of the input points. In testing, we used a regular grid sphere-based approach with the intersection of spheres at 1/3 of their radius and a maximum voting scheme, as in the original paper [25].

## 4. Results and Discussion

To address the problem of the lack of labeled point cloud data for LiDAR data classification for geospatial applications, we developed SP3D—a large-scale hybrid mobile LiDAR point cloud dataset for geospatial applications. Our dataset consists of two subsets: real-world point clouds acquired via Riegl VMX-450 measurement system, and synthetic point clouds generated via the open-source HELIOS++ [58] software. The presence of a universal classification scheme, as well as dense point clouds obtained using a full-fledged mobile mapping system, allows us to use our dataset to study deep learning models in the context of geospatial applications. The presence of both real and synthetic data allows our dataset to be used for domain adaptation.

The results of our work are the following: (1) we developed the dataset, which is described in this section; (2) we evaluated the performance of deep learning models for the semantic segmentation of dense large-scale point clouds in accordance with the proposed classification scheme in the context of geospatial applications; and (3) we assessed the possibility of both real and synthetic data joint use for point cloud representation learning. In this section, we describe the dataset created and the results of corresponding experiments, which we conducted to evaluate the performance of the modern deep learning models for the semantic segmentation of a dense large-scale point cloud trained on our SP3D dataset.

### 4.1. Results of Real-World Data Collection and Annotation

Firstly, we used the KPconv [25] model trained on Paris-Lille-3D [41] dataset to generate initial coarse annotations. Secondly, we used a two-step approach similar to that used in Paris-CARLA-3D [31]. To simplify the further annotation process, the entire point cloud was divided into small tiles 50 by 50 m in size. The patches were then divided between two labelers, and each tile was manually labeled with one of 10 categories. Finally, the labelers exchanged tiles, double-checked all assigned labels, and reassigned them if needed. Manual labeling was performed using the CloudCompare [61] software. The final labeled point cloud is shown in Figure 3.

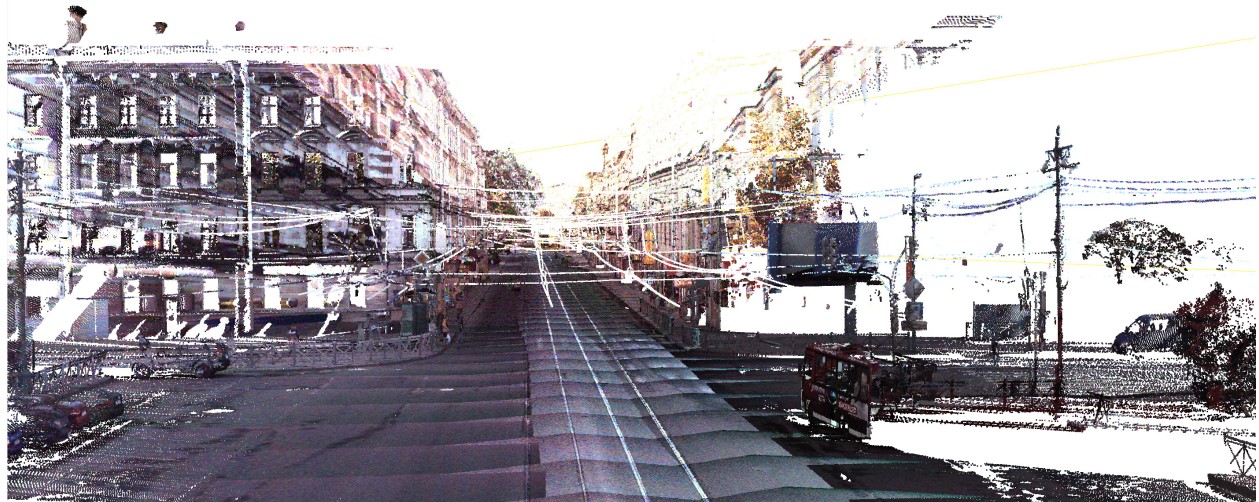

(**a**) Raw real-world point cloud with RGB values

**Figure 3.** *Cont.*

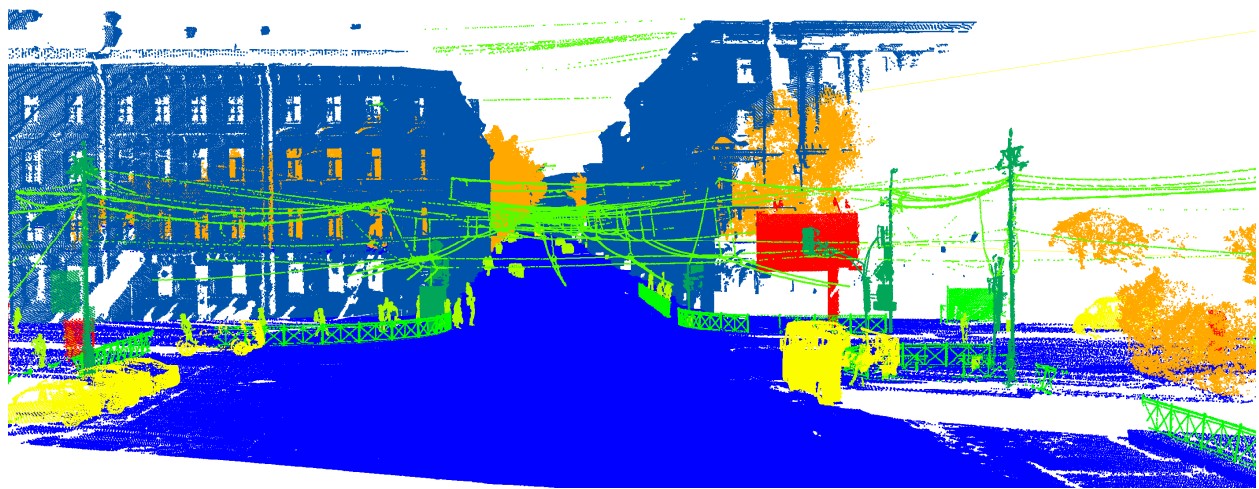

(**b**) Labeled real-world point cloud with 10 categories

**Figure 3.** An example of real-world point cloud data from Komsomola Street.

### 4.2. Results of Synthetic Data Collection

Synthetic data were generated based on standard LiDAR simulation with the sensor's characteristics corresponding to Riegl VQ-450. To run a simulation in HELIOS++, you need to define a survey XML file that contains references to the components needed to build a simulation (the scene, the platform, and the scanner) and the waypoint information needed to define the trajectory of the virtual platform. Firstly, we sorted the 3D objects belonging to the same class into their respective subdirectories (Ground, Building, Pole, etc.). Next, using the Blender2Helios software, we converted the Blender scene to a HELIOS scene [62]. We used a linear motion trajectory, meaning that the waypoint information for all simulations contains two points each (beginning and end), the coordinates of which were determined using the Blender GUI. In our experiment, the values of the scanning parameters were selected in such a way that the distance between points along the scanning line and the distances between the scanning lines of synthetic data coincided with similar distances in real data. We used the following scan parameter values: moving speed = 5.4 m/s, pulseFreq (Pulse Repetition Rate) = 300,000 Hz, and scanFreq (scanlines per second) = 50 Hz.

We generated three synthetic point clouds where each point was assigned one of the eight object categories of proposed classification scheme. There were no points belonging to the categories "Unclassified" and "Artifacts" (explanations are given in Section 4). Each point was also assigned the following seven attributes: heliosAmplitude, hitObjectId, fullwaveIndex, NumberOfReturns, ReturnNumber, GpsTime, and Intensity. Examples of the virtual environment used and corresponding synthetic point clouds are shown in Figure 4.

### 4.3. Dataset Description

Real-world data covers 0.7 km of dense urban area and consists of 34 million points. The synthetic data have a total scanning trajectory length of 2.1 km and consists of 34 million points. Despite the fact that the characteristics of synthetic data (measurement range, density of points, etc.) correspond to the characteristics of real-world data, the number of synthetic points per linear meter of the trajectory of the scanning platform is three times less than in the real scene. This discrepancy is due to the fact that virtual scenes contain fewer objects compared to the real scene.

The original real-world data scene, as well as synthetic scenes, were divided into two parts. For training (56.8% of the total number of points), we used "RealSceneTrain", "SynthScene1Train", "SynthScene2Train", and "SynthScene3Train". For validation (8.2% of the total number of points), we used "SynthScene1Val". Finally, for testing (35% of the total

number of points), we used "RealSceneTest", "SynthScene2Test", and "SynthScene3Test". A summary of the number of labeled points for each class in each file is presented in Table 2.

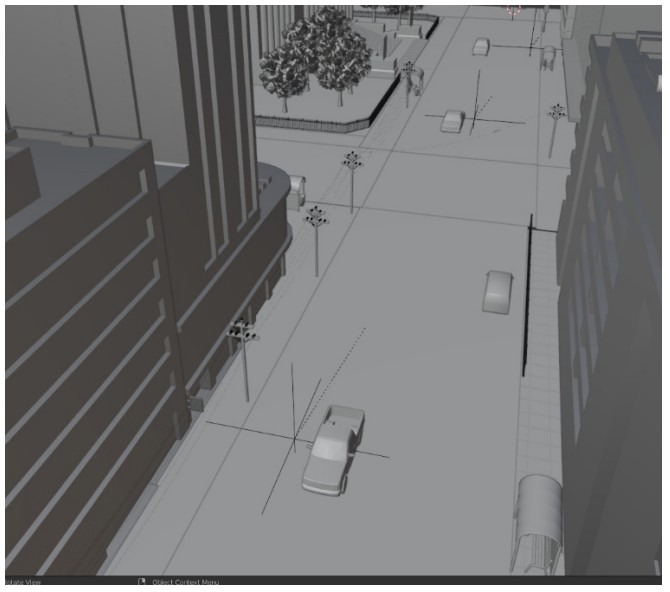

(**a**) Initial 3D model.

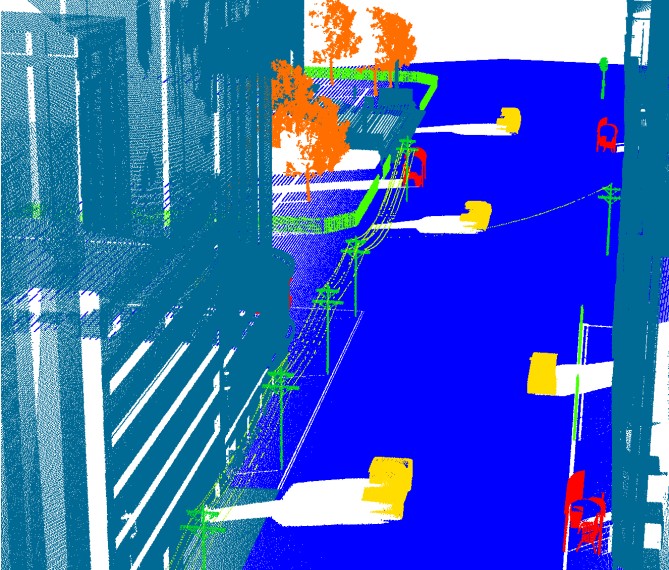

(**b**) Synthetic point cloud generated from it.

**Figure 4.** An example of a virtual environment and corresponding synthetic point cloud data.

**Table 2.** Dataset statistics *.

| File | Ground | Building | Pole | Fence | Utility, Line | Vehicles | Natural | Other, Object | Total |
|---|---|---|---|---|---|---|---|---|---|
| RealSceneTrain | 11,244,113 | 4,885,467 | 66,344 | 178,607 | 129,702 | 127,014 | 2,358,523 | 125,546 | 19,115,316 |
| RealSceneTest | 8,437,117 | 2,283,945 | 34,826 | 1,020,013 | 81,467 | 83,461 | 2,997,606 | 11,435 | 14,949,870 |
| SynthScene1Train | 3,143,519 | 2,999,073 | 21,142 | 166,146 | 19,898 | 191,190 | 57,537 | 44,046 | 6,642,551 |
| SynthScene1Val | 2,640,097 | 2,515,514 | 12,010 | 222,732 | 2668 | 140,816 | 74,447 | 10,365 | 5,618,649 |
| SynthScene2Train | 3,882,534 | 2,992,884 | 13,459 | 45,702 | 4027 | 129,670 | 87,709 | 35,543 | 7,191,528 |
| SynthScene2Test | 3,448,314 | 1,775,707 | 13,164 | 28,648 | 5293 | 34,895 | 40,925 | 20,489 | 5,367,435 |
| SynthScene3Train | 3,586,517 | 2,090,538 | 37,719 | 44,111 | 3300 | 86,690 | 61,758 | 36,732 | 5,947,365 |
| SynthScene3Test | 1,896,448 | 1,407,202 | 26,415 | 90,076 | 1423 | 127,439 | 16,936 | 36,934 | 3,602,873 |

* All values are given in points.

In our experiment, we did not use the "Unclassified" and "Artifacts" classes (explanations are given in Section 3.4.1); thus, in Table 2, we do not specify the number of class data points. At the same time, the real scene of our set contained 4,961 points from the "Unclassified" class and 20,672 points from the "Artifacts" class for future research.

### 4.4. Performance of Baseline Approach

The baseline method trained on proposed dataset performance evaluation results is shown in Table 3. We reached 92.56 mIoU, which demonstrates the high efficiency of using deep learning models for the semantic segmentation of dense large-scale point clouds in accordance with the proposed classification scheme in the context of geospatial applications.

**Table 3.** KP-FCNN [56] model's performance trained on proposed dataset *.

| Scene | mIoU | Ground | Building | Pole | Fence | Utility Lines | Vehicles | Natural | Other Object |
|---|---|---|---|---|---|---|---|---|---|
| RealSceneTest | 87.29 | 99.6 | 94.9 | 65.25 | 88.57 | 94.86 | 97.48 | 98.89 | 58.77 |
| SynthScene2Test | 98.14 | 99.15 | 98.35 | 98.81 | 99.26 | 98.59 | 93.17 | 98.73 | 99.03 |
| SynthScene3Test | 92.25 | 99.4 | 98.8 | 97.5 | 97.6 | 82.3 | 68.3 | 99.9 | 94.2 |
| Mean | 92.56 | 99.39 | 97.35 | 87.19 | 95.14 | 91.92 | 86.32 | 99.17 | 84 |

* All values are given in IoU.

We compared the KP-FCNN's [25] performance using the proposed new dataset to that of other similar datasets. The results of the comparison are shown in Table 4.

**Table 4.** Comparison of KP-FCNN's [25] performances on different datasets *.

| Dataset | Semantic3D [13] | Paris-Lille-3D [41] | SemanticKITTI [42] | Toronto 3D [43] | Urban SGPCM [46] | Paris-CARLA-3D [12] | SP3D (*ours*) |
|---|---|---|---|---|---|---|---|
| KP-FCNN's Performance (mIoU) | 74.6 | 82 | 58.8 | 60.3 | 52.8 | 51.7 | 92.56 |

* All values are given in mIoU.

The achieved performances across different benchmarks exhibit variations, suggesting that the choice of benchmark significantly impacts the performance of the KPConv [25] model. Several factors can contribute to these performance differences:

1. **Dataset Design.** The design of the dataset plays a crucial role in determining the performance of the model. Datasets with a purposeful classification scheme provide a solid foundation for improved model performance.

2. **Annotation Quality.** The quality and consistency of annotations within the benchmark datasets can influence the model's performance. Datasets with more accurate and comprehensive annotations provide better training signals, leading to improved performance.

3. **Class Imbalance.** The presence of class imbalance, where certain object classes are under- or over-represented, can affect the model's performance. Imbalanced datasets may lead to biases in the model's predictions, resulting in lower performance for certain benchmarks.

4. **Dataset Size.** The size of the dataset can also play a role. Larger datasets provide more diverse and representative samples, allowing the model to learn robust features and generalize more effectively. Smaller datasets may suffer from limited coverage and may not capture the full range of variations present in real-world scenes.

Based on the provided performances, it is evident that the SP3D dataset achieved the highest mIoU score of 92.56, outperforming all other benchmarks. This superior performance can be attributed to several factors:

1. **Dataset Design.** The SP3D dataset was purposefully designed to capture the specific characteristics and challenges relevant to the targeted geospatial applications. It included a comprehensive range of object classes, leading to improved model performance.

2. **Annotation Quality.** The SP3D dataset benefited from high-quality annotations, ensuring accurate and detailed labeling of objects. This accuracy enhanced the model's learning process, resulting in higher performance.

The performance variations across benchmarks emphasize the importance of selecting datasets that align with the specific requirements of geospatial applications. The SP3D dataset, which was tailored to the proposed benchmark, demonstrates its effectiveness through achieving higher mIoU scores (92.56). These results underscore the significance of dataset design and quality in enhancing the performance of deep learning models for geospatial tasks. The analysis of achieved performances highlights the impact of different benchmark datasets on the performance of the KPConv model, underscoring the

need for suitable benchmarks that accurately capture the characteristics and challenges of geospatial applications.

## 5. Discussion

The results show that, compared to similar datasets, KP-FCNN's [25] performance using the proposed new dataset is much higher than those of other models.

Analyzing the results of the experiment, we can conclude that for the "RealSceneTest" scene, the most difficult classes in terms of recognition were "Pole" and "Other object". Visualization predictions for the KP-FCNN model on "RealSceneTest" are shown in Figures 5–8.

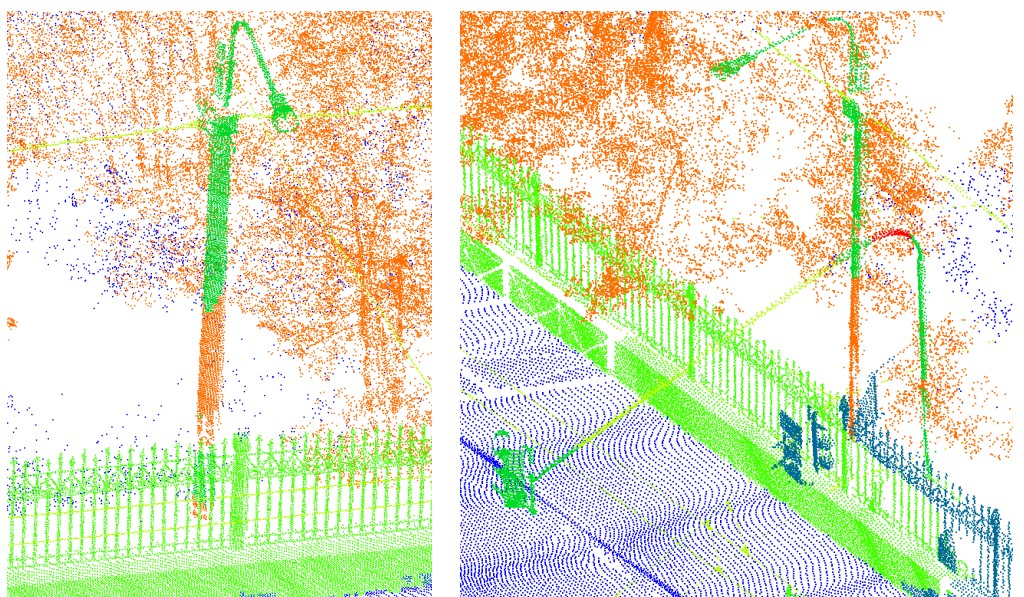

**Figure 5.** Visualization of false negative predictions of points belonging to class "Pole" on "Re-alSceneTest".

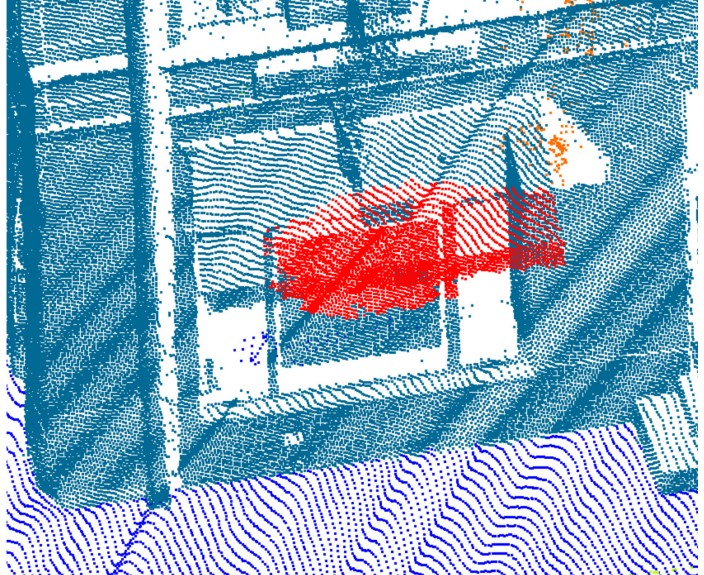

**Figure 6.** Visualization of false positive predictions of points belonging to class "Other object" on "RealSceneTest".

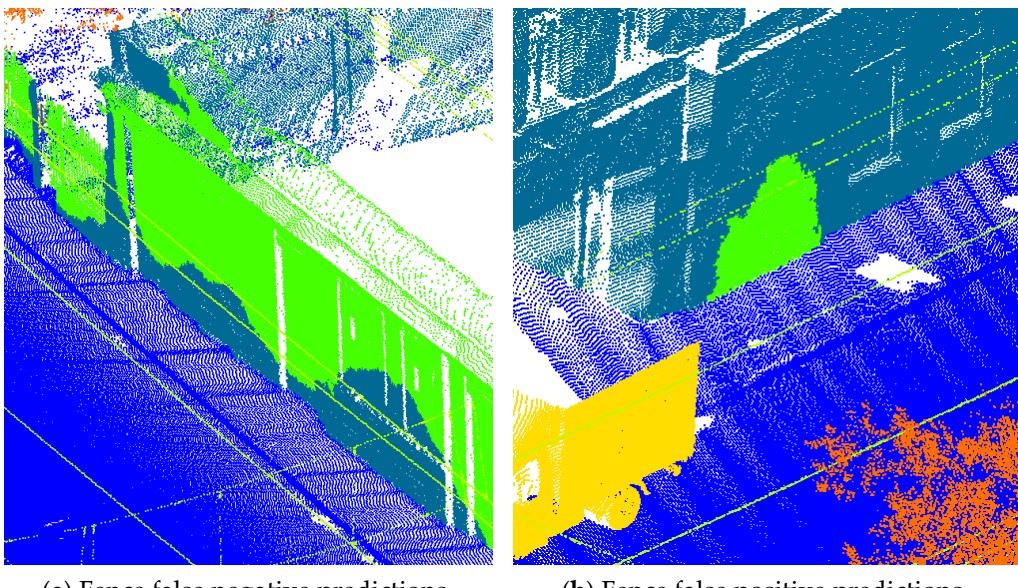

(**a**) Fence false negative predictions  (**b**) Fence false positive predictions

**Figure 7.** Visualization of false negative and false positive predictions of points related to class "Fence" on "RealSceneTest".

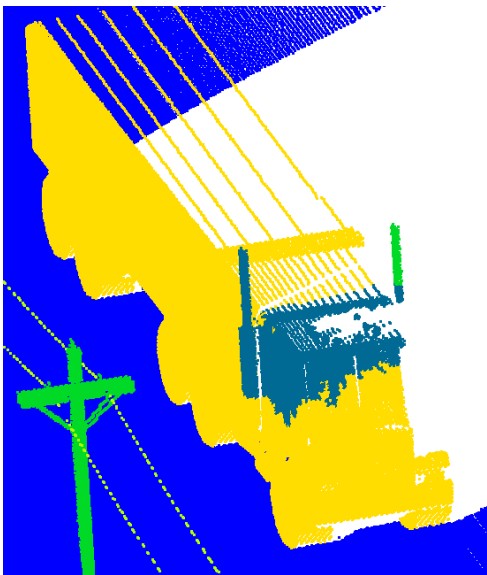

**Figure 8.** Visualization of false negative predictions of points belonging to class "Vehicle" on "SynthScene3Test".

Figure 5 shows false positive predictions of points belonging to the "Pole" class. The figure shows that the model erroneously defines some of the points as vegetation. This result can be explained based on the fact that vertical cylindrical shapes are characteristic of both lighting poles and tree trunks. In addition, the pillars shown in Figure 5 are located in dense vegetation, which affects the contextual information about neighboring points analyzed using the model. Thus, the model "sees" that there are points around the considered point, the distribution of which corresponds to the points of vegetation. We should also take into account the relatively small input sphere radius (3 m). Due to this issue, a subcloud containing only a lighting support is fed to the input of the neural network, and the points of the bracket and lamp are fed into the neural network in another sphere. Thus, the relatively low performance of point recognition for the "Pole" class is associated with the complexity of the objects of this class. One of the most effective ways

to improve the accuracy of point recognition of a given class is to increase the amount of contextual information or, in other words, increase the radius of the original sphere.

We assume that the poor performance of "Other object" recognition is due to the greater variance in instances in this class. On the other hand, according to Table 2, scene "RealSceneTest" contains a small number of points belonging to the class "Other object". Therefore, for this case, misprediction even for one input sphere can lead to a significant decrease in overall performance. Figure 6 shows a part of the facade that was erroneously classified by the model as points of the "Other object" class. The number of false positive points here is around 4000.

Figure 7 demonstrates false negative and false positive predictions for points belonging to the "Fence" class. In Figure 7, we see that the model confuses the points belonging to the "Building" class and the points belonging to the "Fence" class. This problem occurs because the examples of the "Building" and "Fence" classes shown in Figure 7 are flat geometry.

Analyzing the results of the experiment, we can conclude that for synthetic data, the most difficult classes in terms of recognition were "Utility line" and "Vehicles". Prediction result visualization of KP-FCNN model for "SynthScene3Test" is shown in Figure 8.

The low performance of the model for the "Utility line" class on the "SynthScene3Test" is due to the relatively small number of points in this class. Therefore, as in the case of "Other object" for "RealSceneTest", an erroneous prediction for even one input sphere can lead to a significant decrease in overall performance. The low performance of the "Vehicles" class is due to the presence of a truck (see Figure 8) whose cab was mistaken for part of the building.

Analyzing the results, we can conclude that one of the main problems with our point cloud dataset is class imbalance. Artificially augmenting the number of instances of individual objects or using synthetic data with less class imbalance can partially solve this problem; however, this method is uncommon for real-world scenarios. Since low diversity and class imbalance are natural for real-world scenarios, the learning algorithm should be robust to these features [42].

## 6. Future Research Directions

It should be noted that, in our experiment, we do not use any specific features of points, such as RGB or Intensity. Therefore, the use of additional features can significantly improve performance. For example, the experimental results in [12] demonstrate that the use of RGB in addition to geometry improved the overall performance. However, RGB components are the basic elements of images, while point clouds often either contain RGB with inaccuracies and artifacts, or none at all. Intensity is difficult to simulate due to the complex processes of signal transmission, propagation and reception in real conditions [63]. Another promising approach is to use height from the ground value to each point as a feature, since it has strong descriptive power and does not depend on any biases towards synthetic and real data. However, to confirm this assumption, it is necessary to carry out appropriate experiments for geospatial applications.

We believe that in the context of geospatial applications, the domain gap between synthetic and real point clouds depends on two factors: the difference between data characteristics (outliers, removals) and the difference between scene layouts. The results of our experiment showed that when using the baseline method, which includes preliminary optimization of the input point cloud, for scenes with similar layouts, the domain gap between synthetic and real data is insignificant. This gap can be judged based on the high performance of the model trained on the combined dataset. However, despite the fact that we aimed to create 3D models with scene layouts corresponding to the real world, the resulting scene layouts are not identical. Therefore, to unambiguously confirm the possibility of joint use of both real and synthetic data for geospatial applications, additional experiments are required. It is necessary to compare the performance of models when training on real data and when training on synthetic data with identical scene layouts. To do this, you first need to create 3D models based on real data, and then generate synthetic

point clouds. Depending on the results of such an experiment, it will be possible to draw a conclusion about the influence of the characteristics of points on the domain gap.

A promising future research direction is the development of new datasets corresponding to the proposed classification scheme, as well as the reannotation of existing datasets [12,37,41,43–46] in accordance with the developed classification scheme. This approach will help to create a truly large-scale and diverse dataset for geospatial applications. The availability of such a dataset will provide an opportunity to explore modern deep learning methods for the possibility of learning generalized point cloud representations.

Among the promising areas for further research, one should highlight the comparison between the performance of models trained on point clouds and different combinations of features (RGB, Intensity, Height above ground level, etc.). We should also highlight the comparison of the model's performance when training on real data and when training on synthetic data with identical scene layouts. Another promising area of research is the development of methods for the joint use of heuristic algorithms and neural networks to solve the problem of LiDAR data classification for geospatial applications. In the context of geospatial applications, an equally promising direction for further research is the development of multimodal approaches based on the joint processing of raster images and point clouds.

### 7. Conclusions

In this article, we provide an analysis of existing point cloud datasets (point cloud characteristics and classification schemes) in terms of their use in the context of geospatial applications. This analysis showed that they do not fully meet the requirements for spatial data for geospatial applications.

We propose a novel classification scheme that contains object categories most suitable for geospatial applications, instead of the currently used object categories within existing classification schemes, which are more suitable for the perception of the environment using autonomous vehicles. Since our classification scheme contains a set of 10 universal object categories into which any scene represented by dense outdoor mobile LiDAR point clouds can be divided, we recommend using it in the future when developing new point cloud datasets for geospatial applications. Based on the proposed classification scheme, it is possible to develop regulated datasets, which, as a result, can be used as a single dataset for training deep learning models.

We have developed the SP3D dataset—a large-scale hybrid mobile point cloud dataset for semantic segmentation of outdoor scenes for geospatial applications. It contains both real-world (34 million points) and synthetic (34 million points) subsets that were acquired using real and virtual sensors with the same characteristics. Due to the use of a full-fledged mobile mapping platform, the proposed dataset contains more accurate data with a higher density and range than most existing datasets and satisfies the spatial data requirements for geospatial applications.

We presented the results of the performance evaluation of the state-of-the-art deep learning model (KP-FCNN [25]) trained on our dataset. We obtained an overall 92.56% mIoU, which demonstrates the high efficiency of using deep learning models for the semantic segmentation of dense large-scale point clouds in accordance with the proposed classification scheme in the context of geospatial applications. We released our SP3D dataset to stimulate the development of deep learning models for the semantic segmentation of dense, large-scale outdoor mobile LiDAR point clouds for geospatial applications.

**Author Contributions:** Conceptualization, S.L. and V.B.; Data curation, A.C.; Formal analysis, V.B. and A.F.; Funding acquisition, V.B. and A.F.; Investigation, S.L., A.C. and D.Z.; Methodology, S.L. and A.C.; Project administration, V.B. and A.F.; Supervision, V.B. and A.F.; Validation, K.V. and Y.M.; Visualization, A.C. and Y.M.; Writing—original draft, S.L.; Writing—review and editing, V.B. and A.F. All authors have read and agreed to the published version of the manuscript.

**Funding:** The research is partially funded by the Ministry of Science and Higher Education of the Russian Federation as part of the World-Class Research Center Program: Advanced Digital Technologies (contract No. 075-15-2022-311 dated 20 April 2022).

**Data Availability Statement:** The data used in this study are publicly available at: https://github.com/lytkinsa96/Saint-Petersburg-3D-Dataset (accessed on 10 April 2023).

**Conflicts of Interest:** The authors declare no conflict of interest.

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
