# Peer review of "Saint Petersburg 3D: Creating a Large-Scale Hybrid Mobile LiDAR Point Cloud Dataset for Geospatial Applications"

_remotesensing, doi:10.3390/rs15112735_

Round 1

Reviewer 1 Report

This paper presents a mobile LiDAR data benchmark. The authors used KP-FCNN (which is not defined through the manuscript) to classify LiDAR data into eight classes. I do not see any novelty in this article except introducing a new benchmark for mobile LiDAR data. The presented classification scheme that included eight or ten classes are similar to other classification schemes, where the authors combined two or more classes in their scheme. The article is poorly organized and presented. The abstract is very vague. There is no methodology explained in the method section, instead the authors referred to other references. The proposed method achieved a lower performance than existing methods using either machine learning or deep learning. The authors should compare their method with existing methods to justify their method.

Comments:

1-Line 10: driving. But?

2-Line 10: What do you mean by geodetic applications?

3-Line 11: What does it mean “3D LiDAR point cloud machine learning dataset”? You mentioned machine learning and then applied deep learning (Line 18), this is confusing!

4-Line 15: the wording “balanced” means that all classes are represented by equal number of points and that does not match with Table 3?

5-Line 17: “The evaluation protocol for semantic segmentation of point clouds for geodetic applications is described.” What is the protocol?

6-Line 18: What is KP-FCNN?

7-Throughout the introduction, the authors Kept talking about universal classes for Geodetic applications! I reached Page 6 and to find out what is meant by Geodetic applications and their requirements and the 10 universal classes.

8-Line 112: “The proposed dataset contains more accurate data” this claim has to be proven.

9-Line 116: define “KP-FCNN” in full.

10-Table 1: should include at least the accuracy and points density/spacing, these factors that control the data for geodetic applications as stated by the authors!

11-TLS is not defined in the text.

12-Line 128: areal >>aerial

13-The material and methods section does not include any explanation of the deep learning method used?

14-Lines 227-228: no need for these lines

15-Lines 419-427: the data annotation needs more explanation. What the description of KPconv [25] model? What is the two-step approach similar to that used in Paris-CARLA-3D [31]? Illustrations of step-by-step data annotation would be useful

16-Line 468: what is the percentage of training/validation/testing data division? Add the total number of points for each part per each dataset.

17-Table 2: there is no meaning of showing total number of points for each class, mixing real and synthetic data. Instead, add total number of points per each row.

18-Subsection 4.1: Metrics should be moved to Section 3 Methods

19-For the dataset to be acceptable as a benchmark, it should include a wide range of accuracy metrics, such as overall accuracy, kappa, ….

20-Subsection 4.2. Baseline method should be moved to Section 3 Methods

21-Lines 486-491: these sentences are a repetition of Lines 480-485. In addition, the objectives should be stated at the end of the Introduction.

22-Lines 521-529: this paragraph need more explanation. The authors referred to many references without enough explanation.

23-Intersection over Union (IoU) and mean IoU (mIoU), what are the meaning of these two metrics? In other words, what do they represent?

24-Is achieving 87.29 mIoU a good performance? The author should compare the performance of the proposed method with other Deep Learning methods?

25-What is the difference between Figure 3.a and 3.b? Why Figure 3 has two captions a and b, although it shows Pole false negative predictions?

26-Figure 5: same as previous.

27-Line 599: Another promising approach is to use height from

28-Why RGB values weren’t included, although they were collected and processed as mentioned in Lines 407-408?

29-The authors claimed that they proposed a classification scheme of 10 universal categories, but at the end they classified only 8 categories. That made this scheme questionable.

30-The conclusions were not supported by the results.

Moderate editing of English language

Author Response

Dear colleague. Thank you for your comments. Your comments have help us to improve the manuscript. Answers to all comments can be found in the attached file.

Reviewer 2 Report

 This work introduces a new 3D dataset SPB3D: a Large-Scale Hybrid Mobile LiDAR Point Cloud Dataset for Geodetic Applications. The dataset is timely and useful for Deep learning based pointwise semantic segmentation.

From the text, I could not find what does the "SPB" mean. I hope the authors could provide its full name

Author Response

Response to Reviewer 2 Comments

Thank you for your comment: From the text, I could not find what does the "SPB" mean. I hope the authors could provide its full name

 Response: Thank you for your comment. We address it by changing the name of our dataset in order to make it more clear about full name and short abbreviation.

Reviewer 3 Report

This manuscript develops a new large-scale mobile 3D LiDAR point cloud machine learning dataset for the semantic segmentation task of outdoor scenes. The dataset contains real-world (34 million points) and simultaneous (34 million points) subsets for geodetic applications. These subsets are acquired from real and virtual sensors with the same characteristics. A balanced set of 10 generic object classes is included, into which any scene represented by a dense outdoor moving LiDAR point cloud can be classified.

In my opinion, the manuscript begins with an introduction and a second section that introduces and compares several currently dominant point cloud methods, carefully analysing their advantages and disadvantages.j It goes on to propose a different point cloud machine learning method that can be applied to geodetic detection rather than autonomous driving. The manuscript is somewhat innovative, and specific datasets and training and plotting results are also given. In my opinion the presentation quality of the manuscript is up to the standard of remote sensing and could be considered for acceptance with some minor linguistic modifications.

Translated with www.DeepL.com/Translator (free version)

The overall linguistic style of the manuscript is good, with more professional descriptions of scientific vocabulary, but some minor grammatical corrections are needed.

Author Response

Thank you for your comment.

Round 2

Reviewer 1 Report

The authors have improved the manuscript and addressed almost all the comments. The article is now well organized and well presented. Some minor comments:

Point 6: Line 18: define KP-FCNN in full in the abstract.

Point 30: add results values to support the conclusions

In future responses, always double check that you are referring to the correct lines in your responses. The lines you are referring to were shifted in most of responses.

Author Response

Point 6: Line 18: define KP-FCNN in full in the abstract.

Response 6: Thank you, we’ve fixed it (see lines 17-18).

Point 30: add results values to support the conclusions

Response 30: Thank you for this comment. We’ve added results values (see lines 688-689).
